

# Orographically-Induced Spontaneous Imbalance within the Jet Causing a Large Scale Gravity Wave Event

Markus Geldenhuys[1,2], Peter Preusse[1], Isabell Krisch[3], Christoph Zülicke[4], Jörn Ungermann[1], Manfred Ern[1], Felix Friedl-Vallon[5], and Martin Riese[1]

[1]Forschungszentrum Jülich, Institute of Energy- and Climate Research, Stratosphere (IEK-7), Jülich, Germany
[2]South African Weather Service, Private Bag X097, Pretoria 0001, South Africa
[3]Deutsches Zentrum für Luft- und Raumfahrt, Institut für Physik der Atmosphäre, Oberpfaffenhofen, Germany
[4]Leibniz Institute of Atmospheric Physics, University of Rostock, Kühlungsborn, Germany
[5]Karlsruhe Institute of Technology, Institute of Meteorology and Climate Research - Atmospheric Trace Gases and Remote Sensing (IMK-ASF), Karlsruhe, Germany

**Correspondence:** M. Geldenhuys (m.geldenhuys@fz-juelich.de; markusgeld@gmail.com)

**Abstract.** To better understand the impact of gravity waves (GWs) on the middle atmosphere in the current and future climate, it is essential to understand their excitation mechanisms and to quantify their basic properties. Here a new process for GW excitation by orography-jet interaction is discussed. In a case study, we identify the source of a GW observed over Greenland on 10 March 2016 during the POLSTRACC (POLar STRAtosphere in a Changing Climate) aircraft campaign. Measurements were taken with the Gimballed Limb Observer for Radiance Imaging of the Atmosphere (GLORIA) instrument deployed on the High Altitude Long Range (HALO) German research aircraft. The measured infrared limb radiances are converted into a 3D observational temperature field through the use of inverse modelling and limited angle tomography. We observe GWs along a transect through Greenland where the GW packet covers $\approx$ 1/3 of the Greenland mainland. GLORIA observations indicate GWs between 10 and 13 km altitude with a horizontal wavelength of 330 km, a vertical wavelength of 2 km and a large temperature amplitude of 4.5 K. Slanted phase fronts indicate intrinsic propagation against the wind, while the the ground-based propagation is with the wind. The GWs are arrested below a critical layer above the tropospheric jet. Compared to its intrinsic horizontal group velocity ($25-72\,\mathrm{ms}^{-1}$) the GW packet has a slow vertical group velocity of $0.05-0.2\,\mathrm{ms}^{-1}$. This causes the GW packet to propagate long distances while spreading over a large area while remaining constrained to a narrow vertical layer. Not only orography is a plausible source, but also out of balanced winds in a jet exit region and wind shear. To identify the GW source, 3D GLORIA observations are combined with a gravity wave raytracer, ERA5 reanalysis, and high-resolution numerical experiments. In a numerical experiment with a smoothed orography, GW activity is quite weak indicating that the GWs in the realistic orography experiment are due to orography. However, analysis shows that these GWs are not mountain waves. A favourable area for spontaneous GW emission is identified in the jet by the cross-stream ageostrophic wind, which indicates when the flow is out of geostrophic balance. Backwards raytracing experiments trace into the jet and regions where the Coriolis and the pressure gradient forces are out of balance. The difference between the full and a smooth-orography experiment is investigated to reveal the missing connection between orography and the out of balance jet. We find that this is flow over a broad area of elevated terrain which causes compression of air above Greenland. The orography modifies





the wind flow over large horizontal and vertical scales, resulting in out of balance geostrophic components. The out of balance jet then excites GWs in order to bring the flow back into balance. This is the first observational evidence of GW generation by such an orography-jet mechanism.

## 1 Introduction

Gravity waves (GWs) are ever-present in the Earths atmosphere. Gravity waves are emitted in the troposphere by flow over orography (e.g. Bacmeister et al., 1994; Eckermann and Preusse, 1999; Durran, 2003; Geldenhuys et al., 2019), by convection (Pfister et al., 1993; Alexander and Pfister, 1995; Chun and Baik, 1998; McLandress et al., 2000; Beres et al., 2004; Choi et al., 2012; Trinh et al., 2016) and by fronts (Charron and Manzini, 2002). Other sources, such as an out of balance jet (O'Sullivan and Dunkerton, 1995; Zülicke and Peters, 2006; Plougonven and Zhang, 2014), vertical wind shear (Lott, 1997) and a planetary wave induced critical layer in the polar vortex (Polichtchouk and Scott, 2020) occur both in the troposphere and middle atmosphere. The excitation of GWs from an out of balance jet by a geostrophic adjustment frequently occurs during strong Rossby wave activity (Zülicke and Peters, 2006; Plougonven and Zhang, 2014; Ern et al., 2016). Secondary wave generation from breaking GWs (Vadas and Fritts, 2002; Vadas and Becker, 2019; Heale et al., 2020) is another possible source of GWs throughout the atmosphere.

Gravity waves can impact our lives directly through the generation of turbulence endangering air traffic (Fritts and Alexander, 2003; Bramberger et al., 2018; Geldenhuys et al., 2019). Additionally, they are known to enhance and act as a trigger for convection (de la Torre et al., 2011), impact the movement of weather systems and affect the ozone hole (Kidston et al., 2015). Gravity waves are essential drivers of the middle atmosphere circulation (Holton, 2004) through drag deposited by their breaking/saturation (McLandress, 1998; Alexander et al., 2010). By downward coupling, these circulations in the middle atmosphere again impact the surface (e.g. Kidston et al., 2015; Polichtchouk et al., 2018a). Thus, this GW drag must not be neglected. Gravity waves are not properly resolved by most general circulation models (GCMs), hence, GW drag parameterizations are required (Kim et al., 2003; Geller et al., 2013).

General Circulation models use orographic GW drag (OGWD) and non-orographic GW drag (NOGWD) schemes. The OGWD scheme represents the drag exerted by mountain waves alone (Lott and Miller, 1997; Kim and Arakawa, 1995; Xie et al., 2020). The NOGWD scheme is developed to represent all other sources (e.g. Charron and Manzini, 2002; de la Camara et al., 2014a). Parameterization schemes have a number of poorly constrained parameters, and one method of improving models is by finding better constraints by observations (e.g Plougonven et al., 2020).

Direct observational evidence for the relative importance of different GW sources is rare. Hence, often the effect of GWs on the large-scale circulation is used to infer properties of the GW parameterizations (e.g Manzini et al., 1997). A good example is a recent debate on which parameterization scheme is responsible for the missing GW drag around $60°$ S (McLandress et al., 2012). Garcia et al. (2017) suggested that increased orographic sources are the key to solve this missing GW drag problem in models. For this, Garcia et al. (2017) increased the orographic drag for the Southern Hemisphere only. On the other hand,



the European Centre for Medium-Range Weather Forecasts (ECMWF) employed a stronger non-orographic GW drag with favourable results (Polichtchouk et al., 2018b).

Moreover, Charron and Manzini (2002) showed that increased GW emission from fronts provides good results in the Northern Hemisphere, but is less effective in the Southern Hemisphere. Later, Richter et al. (2010) confirmed this by increasing convective and frontal GW sources. Attempts to improve on the realism and to employ physical GW sources (Richter et al.,

2010; Kim et al., 2013) or mimic natural GW intermittency (de la Camara et al., 2014b) are still experimental. Main concerns are that parameterizations use their own assumptions and tunable parameters, which are only weakly constrained by observations. Charron and Manzini (2002), Richter et al. (2010), and Kim et al. (2013) all agree that the trend is toward replacing non-orographic parameterization schemes by source-specific schemes in low resolution models. Richter et al. (2010) continues that GW observations are required to constrain these parameterization schemes.

In particular, the attribution of observed GWs to different sources is in need of further progress. For instance, Hertzog et al. (2008) associated GW momentum flux obtained from superpressure balloon measurements in the Southern Hemisphere polar vortex to orographic and non-orographic sources by regional selection. However, orographic GWs from the Andes and from the Antarctic peninsula can propagate far downstream into the Drake Passage (Rapp et al., 2020). On the other hand, Preusse et al. (2014) and Krisch et al. (2020) show that GWs observed over the Scandinavian mountains may mostly originate from upstream

jet sources. More sophisticated methods than simple spatial collocation are required to identify the sources of observed GWs.

To contribute to this debate, a case over Greenland is discussed in the following. Greenland is an island with a high elevation and surrounded by ocean. This has made it a good place to study wind flow above and around terrain (e.g Doyle and Shapiro, 1999; Tollinger et al., 2019). During this case, a strong Rossby wave is breaking and a GW packet exists over a large part of Greenland. Several of the potential source processes introduced above were present in our case: orography, breaking Rossby

wave, jet streak, as well as strong horizontal and vertical wind shear.

Observations were obtained during the PGGS campaign. The PGGS campaign consisted of smaller sections, namely; POLSTRACC [1], GWEX [2], GW-LCycle and SALSA [3]. One of the major aims of the campaign was the investigation of the generation and life cycle of GWs.

Section 2 describes the data and methods (GLORIA measurements, raytracing, and reanalysis data) as well as the synoptic

conditions. In Sect. 3, a presentation and discussion of the observations follows. Section 3 and 4 contains a discussion on the numerical weather prediction experiment, source identification and GW evolution. The results are summarised in Sect. 5.

---

[1]POLar STRAtosphere in a Changing Climate; Oelhaf et al., 2019

[2]Gravity Wave EXperiment

[3]Seasonality of Air mass transport and origin in the Lowermost Stratosphere and the tropically controlled transition region using the HALO Aircraft



## 2 Data and Methods

### 2.1 GLORIA - Gimballed Limb Observer for Radiance Imaging of the Atmosphere

GLORIA (Friedl-Vallon et al., 2014; Riese et al., 2014) is an imaging infrared spectrometer which is mounted in the belly pod

of HALO (the German High Altitude Long Range research aircraft). The instrument comprises a Michelson interferometer with a 2D infrared detector array. GLORIA looks to the right side of HALO with regards to flight direction and its field of view can be panned from $135°$ to $45°$ w.r.t. carrier heading in the horizontal. The vertical field of view is $4.1°$. With this, we image altitudes from $\sim 5\,\text{km}$ to slightly above flight altitude.

GLORIA measures spectra between 780 to $1400\,\text{cm}^{-1}$. This allows measurement and retrieval of temperature, $O_3$, $H_2O$,

$NH_3$, PAN, $ClONO_2$ and $HNO_3$. GLORIA uses $48 \times 128$ pixels of the detector to provide $\approx 6000$ simultaneous views. Each pixel is analysed for absorption lines of the above-mentioned gasses (Friedl-Vallon et al., 2014).

GLORIA can measure with a spectral sampling of up to $0.0625\,\text{cm}^{-1}$. However, the finer the spectral sampling, the longer is the acquisition time needed to achieve this. A longer integration time implies a worse spatial resolution, as the aircraft is constantly moving. A lower integration time allows a finer spatial resolution, but impacts on the number of trace species that

can be retrieved (Friedl-Vallon et al., 2014; Riese et al., 2014; Ungermann et al., 2010a). Based on the integration time, three main observation modes exist, Chemistry mode, Dynamics mode and Premier mode. Chemistry mode uses a spectral sampling of $0.0625\,\text{cm}^{-1}$ for an increased number of detectable chemical species. Dynamics mode uses $0.625\,\text{cm}^{-1}$ for an increased spatial resolution and to focus more on the retrieval of atmospheric temperature. Intermediate Premier mode employs a value of $0.2\,\text{cm}^{-1}$ as a compromise.

On 10 March 2016, GLORIA was flown in Dynamics mode with the aim to perform tomographic retrievals. Tomographic measurements utilise the panning ability of GLORIA. During the flight, GLORIA was panned from $129°$ to $45°$ in steps of $4°$. This provides multiple measurements of the same air mass from different angles. Data from different angles allow for a tomographic retrieval with the help of the GloriPy (Kleinert et al., 2014) and JURASSIC2 (Juelich Rapid Spectral Simulation Code version 2; Ungermann et al., 2010b) software packages. Reconstructing the atmosphere from infrared observations is an

ill-posed inverse problem. To solve this problem, an atmospheric state is iteratively adjusted by a Gauss-Newton type trust-region method (Ungermann, 2011). This continues until the synthetic measurements generated by a forward model agree within expectation to the actual measurements. The final state of this iterative process is then used as the 'retrieval' result (Krisch et al., 2017; Krasauskas et al., 2019).

The retrieval data generated for this article was optimised to determine temperature, $CCl_4$, $HNO_3$, $O_3$ and aerosols. The

spectral windows used for the optimised retrieval are listed in Appendix A. Sixteen channels were used in the retrieval, each for a different purpose (3 for temperature, 5 for $CCl_4$, 4 for $HNO_3$ and 4 for temperature and $O_3$ combined). The retrieval was conducted using Laplacian regularisation and an irregular grid (Delaunay method) (Krasauskas et al., 2019). The Laplacian (second-order derivative) regularisation replaced the traditional first spatial derivative regularisation approach. The Delaunay grid reduces the computational expensiveness of the tomographic retrieval. This has been the very first irregular grid GLORIA

retrieval.



Tomography employs multiple views from different angles for examining a given target area. During full angle tomography, the aircraft follows a closed (e.g., circular or hexagonal) flight path around the area of interest. During limited angle tomography (LAT, used to obtain the data for this article) the aircraft flies in a (largely) straight line. During linear flight patterns, the area of interest is observed from fewer angles (Krisch et al., 2018, 2020). Fewer angles mean more difficulty in 3D retrieval and

frequently more artefacts. The resolution is also slightly worse during LAT. The resolution of our LAT is $200\,\mathrm{m}$ in the vertical and $\sim$20-70 km in the horizontal direction.

To examine the robustness of our results, we tested different retrieval configurations. We found the derived temperature product to be robust within the region of high tangent point[4] density, whereas other parts of the volume were subject to large differences depending on the chosen a priori or regularisation. The a priori used in the retrieval is a smoothed ECMWF analysis

and WACCM (Whole Atmosphere Community Climate Model) reanalysis field.

## 2.2   GROGRAT - Gravity-wave Regional Or Global Ray Tracer

GROGRAT is a raytracing tool that traces the propagation path of a GW and can be used for both forward and backward tracing (Marks and Eckermann, 1995; Eckermann and Marks, 1997). GROGRAT is based on the GW dispersion relation:

$$\omega^2 = \frac{(k^2 + l^2)N^2 + f^2(m^2 + \frac{1}{4H^2})}{k^2 + l^2 + m^2 + \frac{1}{4H^2}} \tag{1}$$

where $\omega$ is intrinsic frequency, $N$ is Brunt-Vaisala frequency, $f$ is Coriolis frequency, $H$ is scale height and $k, l, m$ are wavenumber in $x, y, z$ direction. A GW packet is fully characterised by its position in space and time and its 3D wave vector. The ray-tracer projects this state vector forward or backward according to the ray-tracing equations (Lighthill, 1978):

$$\frac{\partial x_i}{\partial t} = \frac{\partial \omega}{\partial k_i} \tag{2}$$

$$\frac{\partial k_i}{\partial t} = -\frac{\partial \omega}{\partial x_i} \tag{3}$$

where $i$ denotes the spatial direction ($x, y,$ or $z$), and $\frac{\partial}{\partial t}$ is differentiation in time.

In this study, the 4D version of GROGRAT is used. This means that the background (see next section to see how the background state was determined) temperature, wind and pressure (from ERA5 reanalysis) change with time. For each time step the group velocity and state vector ($\omega_{groundbased}, k, l$) change. Along the ray path, wave action density ($A \equiv \frac{\bar{E}}{\omega}$)[5] is conserved, but GW saturation as well as GW dissipation by radiative damping and turbulence are taken into account. Gravity

wave amplitudes are converted from wave action.

---

[4]A tangent point is a point along the line-of-sight which is closest to the earth surface. This point marks the part of the line-of-sight with the largest atmospheric density where usually most of the radiance signal comes from.

[5]where $\bar{E} = \frac{1}{2}\rho(\frac{\hat{T}}{T})^2(\frac{g}{N})^2\frac{\omega^2}{\omega^2 - f^2}$, here $\hat{T}$ is temperature amplitude and $T$ is temperature





For backtracing, it is important to keep in mind that the GW can be emitted at any point along the ray, and is not necessarily emitted at the lowest point of the ray (cf. Preusse et al., 2014). One indication of a GW source along the ray is a violation of the Wentzel-Kramers-Brillouin (WKB) approximation (Hertzog et al., 2001). The WKB approximation requests that the scales of change of the background are large compared to the wavelength of the GWs. This is tested via the parameter (cf. Eq. 5 of

Marks and Eckermann (1995))

$$\delta = \frac{1}{m^2}\left|\frac{dm}{dz}\right| \tag{4}$$

## 2.3 Reanalysis data and model integrations

Throughout this study (except Sect. 4) we use ERA5 (European Centre for Medium-Range Weather Forecasts Reanalysis 5th Generation Description; Hersbach et al., 2020) reanalysis. The ERA5 data are interpolated to a $0.3° \times 0.3° \times 200\,\text{m}$ grid.

All data used in this article is on a geopotential height grid (with exception of Fig. 1 and the calculation of the cross-stream ageostrophic wind (Sect. 4), which is calculated on a pressure grid).

To investigate the influence of orography, in Sec. 4 two global model forecasts with ECMWF Ingerated Forecast System (IFS) are discussed: i) CTL-run and ii) T21-run. The forecasts are perfomed at TCo1279 horizontal resolution (corresponding to 9km grid-spacing on a cubic octahedral grid) with 137 vertical levels and use the operational ECMWF IFS configuration of cy-

cle 45r1 (https://www.ecmwf.int/en/forecasts/documentation-and-support/evolution-ifs/cycles/summary-cycle-45r1). The only difference in the two runs is the resolved orography field, which in CTL-run is at TCo1279 resolution and in T21-run at T21 resolution. This mean that the orography in T21-run is much smoother, does not resolve, for instance, Fjords at the Greenland coast and is only 60% of the TCo1279 orography field elevation. The two forecasts were initialised on 9 March 2016 at 12:00 UTC and run freely for $72\,\text{h}$ (the GW observation takes place $30\,\text{h}$ after initialisation).

The model output and reanalysis data were separated into GWs and large-scale background state. Zonally the data were separated with a Fast Fourier Transform, assuming that zonal wavenumbers up to 12 can be attributed to the large scale background and that higher zonal wavenumbers are attributed to GWs (Strube et al., 2020). In the two remaining directions, a Savitzky-Golay filter was applied. A third-order polynomial was applied in the $y$ (across-latitude) direction with a 50 point smoothing — $15°$ of latitude. A fourth-order polynomial was applied in the $z$ (vertical) direction with a 15 point smoothing —

$3\,\text{km}$. The GW field (called the residual) results after subtracting the large scale background from the original model field. A comparison of different background removal methods can be found in the appendix of Krisch et al. (2020).

## 2.4 Jet geostrophic balance calculation

A jet can generate GWs, if an imbalance exists between the Coriolis and the pressure gradient forces in the momentum equation (Zülicke and Peters, 2006). The area of imbalance normally occurs in the jet exit region and can radiate GWs spontaneously

in an attempt to balance the Coriolis and pressure gradient forces. In this article, the cross-stream ageostrophic wind is used to diagnose unbalanced flow within the jet (similar to Zülicke and Peters, 2006; Mirzaei et al., 2014). First, horizontal wind





($u$ and $v$ components) and geopotential height fields were smoothed to remove GW signatures with a boxcar over $500\,\mathrm{km}$ in $x$ and $y$ direction (similar to Mirzaei et al. (2014)). [6] Then, ageostrophic winds ($u_a$ and $v_a$) were calculated on pressure level data, using MetPy (May et al., 2008 - 2020). Finally, the cross-stream ageostrophic wind was calculated using the approach of Zülicke and Peters (2006):

$$U_c = \frac{u_a v - v_a u}{(u^2 + v^2)^{1/2}} \tag{5}$$

## 3 Observations and GW raytracing

### 3.1 Synoptic situation

The synoptic situation for our case study is shown in Fig. 1. The meandering $300\,\mathrm{hPa}$ geopotential height and horizontal wind field show a cyclonically breaking Rossby wave. At flight time (10 March 18:00 UTC, panel b), the potential vorticity lines steepen and turn back at the point of inflection, signalling Rossby wave breaking. An associated mid-tropospheric low-pressure system drifts from west to east (not shown). Accordingly, the sub-tropical jet drifts with time. However, the divergence of the jet remains above or in close vicinity of Greenland throughout the $30\,\mathrm{h}$ before observation. The above mentioned synoptic conditions are favourable for the formation of jet generated GWs (e.g. Uccellini and Koch, 1987; Plougonven and Zhang, 2014).

A trapping layer inhibits GW propagation beyond the the respective layer. A trapping/reflection layer is formed by an increase in wind speed and stability and is identified with the help of the Scorer parameter (e.g. Durran, 2003; Geldenhuys et al., 2019). Knowing potential GW reflection layers can therefore be important to find the sources of GWs. The Scorer parameter over the southern mainland of Greenland (not shown) indicates multiple reflection layers between $7.5\,\mathrm{km}$ and $13\,\mathrm{km}$. However, further investigation reveals that up to $30\,\mathrm{h}$ before observation all vertical wavelengths $> 4\,\mathrm{km}$ will be unaffected by these reflecting layers. With no reflecting layers present for the wavelengths considered in this study, it is clear the GW source can be located at the surface, or in the free troposphere. Furthermore, this justifies the use of ray-tracing tools for freely propagating GWs (Sect. 3.3).

A jet is known to release upward (above the jet) and downward (below the jet) propagating GWs (Thomas et al., 1999; Guest et al., 2000). Hodographs can be used to distinguish between upward and downward propagating GWs. In the Northern Hemisphere clockwise (anti-clockwise) rotating hodographs indicate upward (downward) propagating GWs (Andrews et al., 1987; Hertzog et al., 2001). Multiple hodographs from ERA5 reanalysis were drawn within the 500 and $300\,\mathrm{hPa}$ jet region. The hodographs (e.g. Fig. 2) depicted no rotation to weak clockwise rotation with altitude below $7\,\mathrm{km}$, with strong clockwise rotation above $10\,\mathrm{km}$. The rotation above $10\,\mathrm{km}$ is nearly circular, which implies that $f/\omega \approx 1$. This points to an upward propagating inertia-gravity wave with a low intrinsic frequency close to the Coriolis parameter (Hertzog et al., 2001; Fritts

---

[6]Zülicke and Peters (2006) smoothed over $1000\,\mathrm{km}$, but for this case smoothing over $500\,\mathrm{km}$ better conserved the synoptic wind structure and was sufficient to remove all GWs.





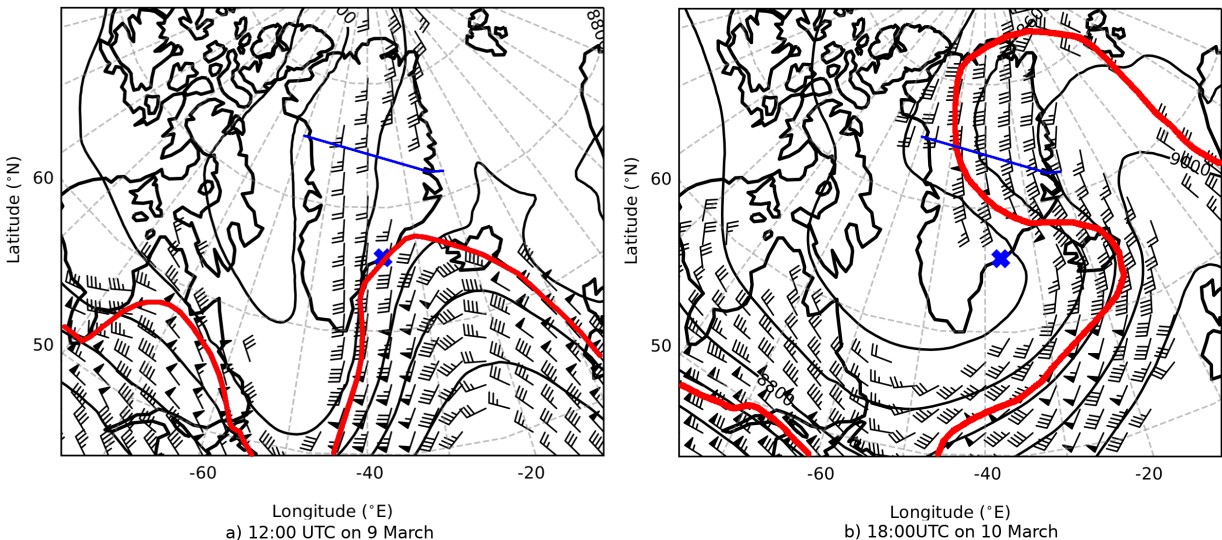

**Figure 1.** ERA5 geopotential heights and winds at $300\,\text{hPa}$ valid for 9 March 12:00 UTC (a) and 10 March 18:00 UTC (b). Only wind speeds greater than $20\,\text{ms}^{-1}$ are plotted. One full barb represents $10\,\text{ms}^{-1}$ and half a barb $5\,\text{ms}^{-1}$. The red line represents the approximate (excluding eddies and local fluctuations) $4\,\text{PVU}$ line. The blue cross will be referred to in Sect. 3.3. The blue solid line represents the flight section used for GLORIA retrieval.

and Alexander, 2003). Less pronounced is nearly a full anticlockwise rotation between 7 and $10\,\text{km}$, even though this altitude range should be treated with care as the jet region (7.5 to $10\,\text{km}$ at flight time) can present artificial results.

Two days before the flight was actually performed, high-resolution ECWMF medium-range weather forecasts predicted a large-scale GW event covering most of Greenland. Accordingly, a PGGS research flight was planned to measure these GWs, presumably generated by the breaking Rossby wave. HALO flew from Kiruna, Sweden to Greenland where it crossed the mainland from south-east to north-west at $10.5\,\text{km}$, and, on the way back at $13.5\,\text{km}$. The temperature field presented in this article is retrieved from this higher leg (black line crossing Greenland in Fig. 3) from 19:00 UTC (Universal Time Coordinated) to 21:00 UTC. Throughout this article, the closest synoptic time (18:00 UTC) is referred to as flight time.

### 3.2 GLORIA observations

Gravity waves are seen within the tangent point area in Fig. 3. Outside this area the retrieval does not have measurement information and falls back to the a priori. The GWs within the tangent point area compare well to ERA5 data (more on this in Sect. 3.3). In the horizontal (panel a), the GW phase fronts are oriented at about $90°$ angle to the flight path. With height (panel b), the phase fronts slant eastwards. Slanting GW phase fronts are an indication of vertically propagating, internal GWs. The observed slant together with the hodograph analysis indicate upward propagation, with intrinsic propagation to the east.





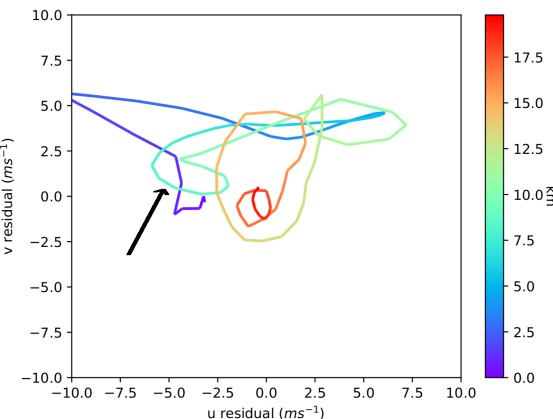

**Figure 2.** Hodograph in the centre of the jet upwind of Greenland at 60°N and -30°E. The arrow points to the anti-clockwise rotation between 7 and 10 km, with the rest being clockwise or no rotation. The hodograph is valid for 9 March at 07:00 UTC, calculated from ERA5, and a good representative for all hodographs upwind, within the jet and through time.

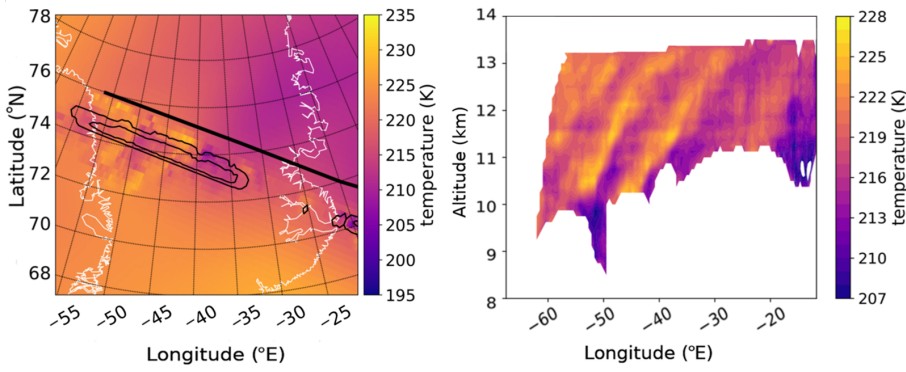

**Figure 3.** Horizontal (left) and vertical (right) cross-section of the GLORIA temperature retrieval, showing the retrieved GW packet. The horizontal cross-section is shown for 11 km. The white line represents the Greenland coastline. The thick black line represents the section of the flight path that was used for the retrieval, the thin black contours represent the tangent point area. The vertical cross-section is averaged at 90° to the flight path using only data within the tangent point area. Note the decrease in vertical wavelength and amplitude above 12 km.





**Table 1.** GW characteristics determined by eye from the retrieval (Fig. 3). The horizontal wavelengths are represented by $\lambda_x$ and $\lambda_y$ for the x and y direction respectively. Ray 0 to 3 were used as input for the GROGRAT raytracer.

| GW number | Lat (° N)/ Lon (° E) | Alt (km) | $\lambda_x$ (km) | $\lambda_y$ (km) | Groundbased frequency ($s^{-1}$) | Temp amplitude (K) |
|---|---|---|---|---|---|---|
| 0 | 74.3/-38.8 | 12.3 | 386.2 | 1.6 | 9.6e-5 | 3 |
| 1 | 73.9/-41.5 | 12.0 | 320 | 2.0 | 1.2e-4 | 4.5 |
| 2 | 74.5/-43.3 | 11.4 | 335.8 | 2.0 | 1.3e-4 | 4.5 |
| 3 | 74.0/-45.0 | 11.0 | 330.1 | 2.1 | 1.6e-4 | 4.5 |

The GW characteristics are determined within the tangent point area indicated in Fig. 3 and similar retrieved images. The characteristics of these GWs are listed in Table 1 and was used as input into GROGRAT (see next section). A horizontal wavelength between 320 km and 390 km is observed in different parts of the GW packet. The vertical wavelength is between 1.6 and 2.1 km, and the GW orientation between 130° and 140° from North. The amplitude and vertical wavelength decrease with altitude (as can be seen in Fig. 3b), despite the expected increase of amplitude with a decrease in density. This is indicative

that a change in propagation conditions is taking place and can point to GW dissipation (more on this in Sect. 3.3).

### 3.3 GROGRAT raytracing

Tracing the backward trajectory of a GW is an established method to find its source (e.g. Marks and Eckermann, 1995; Krisch et al., 2017, 2020). According to Hertzog et al. (2001), the excitation of GWs by geostrophic adjustment from the jet is usually associated with enhanced values of the WKB parameter ($\delta$) near the height of the wind maximum. This is attributed to the

sharp upper and lower edges of the jet.

    Four main rays were backtraced, starting between 11 and 12.3 km based on the GW parameters given in Tab. 1. The GW ground-based frequency or input to GROGRAT was calculated via the dispersion relation (Eq. (1)) using the horizontal- and vertical wavelength in Table 1 as well as ERA5 reanalysis data.

    All rays trace backward into the jet and end over the ocean, with exception of ray 3 (Fig. 4; rays named as in Tab. 1). As

the vertical cross-section of the GLORIA observation indicates the GW is propagating intrinsically opposite to the wind. The horizontal group velocity, however, is slower than the wind velocity which leads to a downstream drift of the GW packet. To provide further confidence in the raytracing study, sensitivity tests were performed. An ensemble raytrace (20 members) was conducted by perturbing the initial conditions (listed in Tab. 1) by $\approx$10%; $\pm$0.2 km for the vertical wavelength and $\pm$30 km for the horizontal wavelength. This is the approximate error associated with the wavelength determination from Fig. 3. All

ensemble members behaved similarly to the main rays (Fig. 4). The ray paths proved to be more sensitive to the launch orientation. A 10° change in orientation frequently ended with the backtraced GW being evanescent or vertically stalling. In





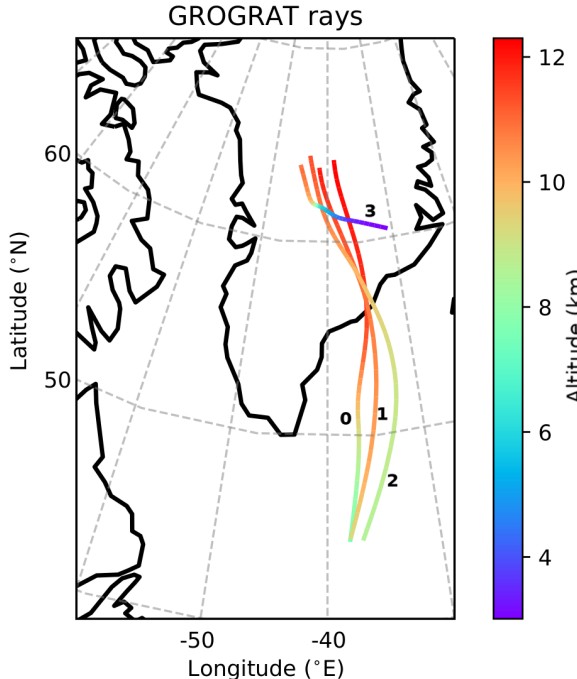

**Figure 4.** GROGRAT backtracing rays, using a 4D setup. The rays have a starting time of 10 March 2016 at 18:00 UTC. The initial conditions to the rays are listed in Table 1. The backtracing starts between 11 and 12.3 km and is depicted respective to their altitude, latitude and longitude.

another experiment, the 4 main rays were backtraced, whereby the ray orientation at the end of the ray was perturbed (again by 10°) and forward traced. In the forward tracing, a change in orientation was much less sensitive.

Ray 0 to 2 (named as in Tab. 1) all experienced large horizontal propagation and very little vertical propagation. This is
normally characteristic of trapped GWs, however, the slanted phase fronts in Fig. 3 indicates the GWs were not trapped. Only ray 2 is discussed here in detail (Fig. 5). The wavelengths and phase orientation predicted by GROGRAT correlates well with the ERA5 reanalysis and produces further trust in the experiment (same as for ray 3 - Fig. 6). The GROGRAT-calculated vertical group velocity peaks at 10 km with $0.2\,\mathrm{ms}^{-1}$ and has a minimum of $0.05\,\mathrm{ms}^{-1}$ at 9 and 11 km (Fig. 7). This translates to a vertical propagation speed of 180 to $720\,\mathrm{mh}^{-1}$. Intrinsic horizontal group velocity peaks at $72\,\mathrm{ms}^{-1}$ and has a minimum
of $26\,\mathrm{ms}^{-1}$. This translates to a ground-based group velocity ranging from $6\,\mathrm{ms}^{-1}$ to $17\,\mathrm{ms}^{-1}$.

The cross-stream ageostrophic wind (calculated as described in Sect. 2.3) indicates out of geostrophic balance flow within the jet exit region at multiple locations along the ray path (Fig. 5). In this study, we use a safe value of $7.5\,\mathrm{ms}^{-1}$ to indicate that the jet exit region is out of balance and can spontaneously emit GWs. Mirzaei et al. (2014) used $1\,\mathrm{ms}^{-1}$ to indicate out of balance areas in the jet, and argued theoretically $4\,\mathrm{ms}^{-1}$ are a good value. On Fig. 5 at x=-1500 km the ray passes through





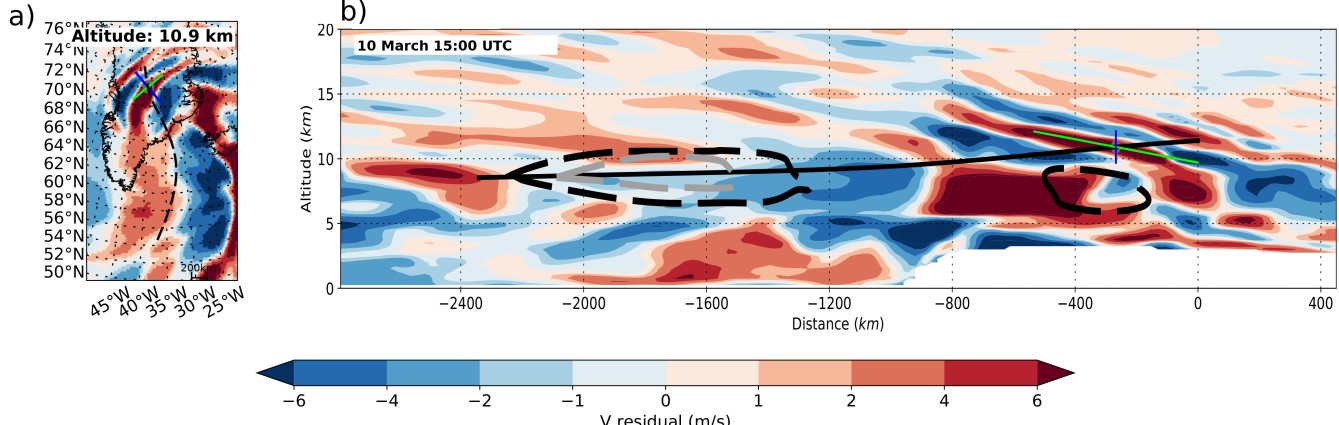

**Figure 5.** GROGRAT backtracing of ray 2 and meridional wind ($v$) residuals from ERA5. The backtracing starts at 11.4 km (on 10 March at 18:00 UTC) and ends at 8.5 km (on 9 March at 18:00 UTC). Panel a shows the horizontal cross-section of the ray path (dashed black line) and the corresponding $v$ residual wind speeds at 10.9 km. Panel b shows the vertical cross-section of $v$ residual wind speeds along the ray path (solid black line). Raytracing started at ray reference time 42 h (flight time), both a and b are valid for 13:00 UTC (5 h prior to ray initialisation). The cross of the blue and green line represent the position of the GW at the respective time, the blue line indicates one horizontal (vertical) wavelength in the horizontal (vertical) cross-section and the green line indicates the orientation of the phase fronts, all calculated in GROGRAT. The thick dashed black (grey) line indicates cross-stream ageostrophic wind values in excess of $7.5\,\mathrm{ms}^{-1}$ ($10\,\mathrm{ms}^{-1}$). The encircled dashed area at -1500 km occurred between 24-19 h before ray initialisation (the backtrace passed through this area at the same time). The encircled dashed area at -400 km represents an out of balance jet at 2-7 h before ray initialisation.

a $10\,\mathrm{ms}^{-1}$ cross-stream ageostrophic wind region 22 h before observation. Multiple other out of balance regions exist within the jet throughout the ray lifetime. From this, it is concluded that the jet is constantly emitting GWs. It is noted that ray 0 to 2 did not have major WKB violations. Although WKB values reached a maximum within the out of geostrophic balanced jet regions, the peak values reached mere values of 0.5.

Ray 3 is the only GW which traces to the orography (Fig. 6) and hence was investigated for a possible mountain wave. The ray traces to the plateau of Greenland, and not the precipitous coastline orography. In addition, the ray passes through a cross-stream ageostrophic wind region at the steepest part of the ray. Clear violations are observed in the WKB values at 6.5 and 8 km (consistent with the findings of Hertzog et al. (2001)), supporting the idea that the GW is released by the jet. It should be noted that cross-sections through ERA5 reanalysis residual data did indicate mountain wave activity localised in time and space along the coast. However, as far as the GROGRAT experiment is concerned, these mountain waves were not observed by GLORIA. Ray 3 was emitted by the jet initially with a longer vertical wavelength (hence the steep propagation path at $x = -250$ km), which was immediately reduced as it propagated out of the strong wind regime. The vertical group velocity peaked around $8\,\mathrm{ms}^{-1}$ (Fig. 7), receding rapidly to $0.04\,\mathrm{ms}^{-1}$ at 11 km.

The vertical wavelength of rays 0 to 2 similarly decreased as the GW passed above the jet region (Fig. 7 left). The effect of the changing vertical wavelength is also observed on the vertical group velocity (Fig. 7 right). Similarly the observed GW





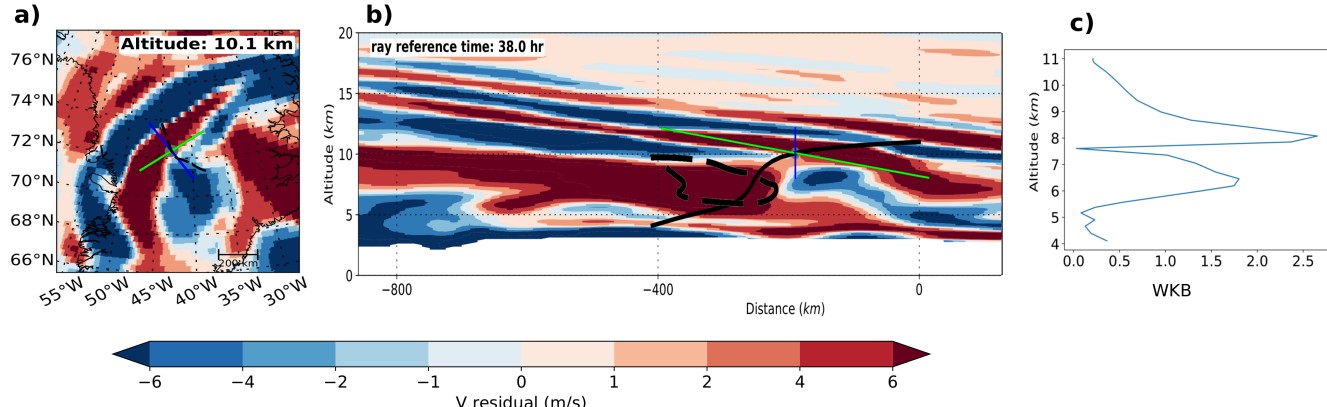

**Figure 6.** Similar (a and b) as in Fig. 5, but valid for ray 3 on 10 March at 14:00 UTC (4 h prior to ray initialisation). The horizontal cross-section (panel a) is valid for 10.1 km. Panel b shows the raytracing starts at 11 km and ends above the Greenland plateau. The thick dashed black (grey) line indicates cross-stream ageostrophic wind values in excess of $7.5\,\mathrm{m\,s^{-1}}$ ($10\,\mathrm{m\,s^{-1}}$). Panel c indicates the WKB parameter with height. The relation of the WKB parameter can be related to the jet in Fig. 8.

amplitude and the observed vertical wavelength decrease with height (Fig. 3 and Tab. 1), this can imply that the GWs are approaching dissipation. A large drop in background wind speeds ($17\,\mathrm{m\,s^{-1}}$ for ray 3 and $40\,\mathrm{m\,s^{-1}}$ for ray 0 to 2) occur from 8 to 9 km to the ray starting altitude (Fig. 8). Similarly the stability ($\frac{\partial T}{\partial z}$) of the atmosphere within the jet exit region (location indicated by the blue cross on Fig. 1) changes significantly, from $0.00025\,\mathrm{K/100m}$ between 8 km and 12 km to $0.2\,\mathrm{K/100m}$ between 12.5 km and 15 km. The strong decrease in wind speed and stability is responsible for the decrease in amplitude and

vertical wavelength, and hence responsible for the GW dissipation. As the wind speed approaches the horizontal phase velocity, the intrinsic frequency decreases to zero which means the vertical wavenumber will approach infinity; representing a critical layer for the GW.

## 4 Numerical Weather Prediction Experiment and GW source identification

### 4.1 Numerical experiment overview and results

The unmodified ECMWF operational model was used as the control (CTL-run 2.3) (Fig. 9a-h). The CTL-run produced GWs (Fig. 9e) similar to observations (Fig. 3) and ERA5 data (Figs. 5 and 6).

The second experiment (T21-run) uses a T21 topographic field (the lowest resolution topographical field available), to achieve a smoothed orography. Comparing the CTL-run with the T21-run on 10 March at 18:00 UTC (Fig. 9e and m), in the area of interest GWs are observed in the CTL-run, but hardly any are seen in the T21-run (also seen in the following time





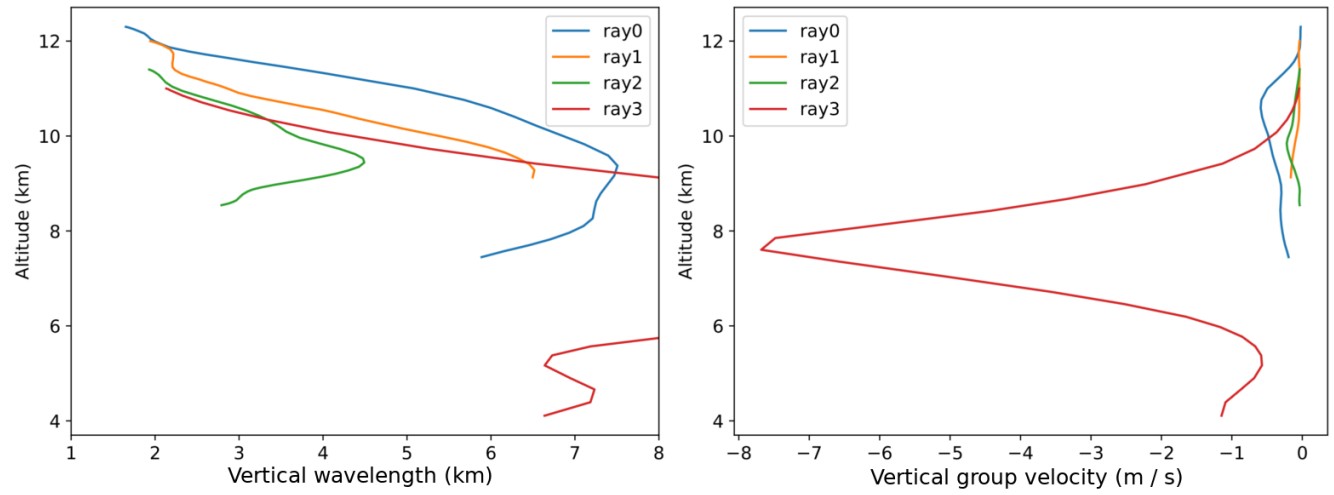

**Figure 7.** Vertical wavelength (left) and vertical group velocity (right) along the backtrace for ray 0 to 3 as calculated by GROGRAT. The leftmost plot is cut at 8 km in order to achieve readability for ray 0 to 2.

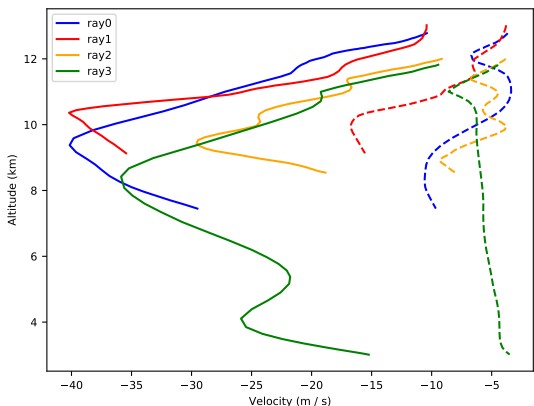

**Figure 8.** The horizontal phase velocity (dashed) and the background wind (solid) along the rays 0 to 3. Where the phase velocity and the wind speed approach one another a critical layer exists. Altitudes above the ray starting point (Tab. 1) represents the same starting conditions raytraced forward.







**Figure 9.** CTL-run (a–h) and T21-run (i–p) cross-stream ageostrophic wind and temperature residuals at different times. The cross-stream ageostrophic wind (a–d and i–l) was calculated on pressure levels, hence here it is depicted at 350 hPa (≈8.1 km). The temperature residuals (e–h and m–p) were determined on geopotential heights, hence this is valid for 10 km. The temperature residuals are depicted ≈2 km higher than the cross-stream ageostrophic wind, as the GW structure forms a complex interference pattern with upward and downward propagating GWs within the jet. The temperature residual plots are offset by 6 h from the cross-stream ageostrophic wind to allow time for the GWs to propagate to 10 km (a vertical group velocity of $200 - 700\,\mathrm{m\,h^{-1}}$ indicates vertical propagation of 2km between 3 to 10hrs). The overlaid wind barbs are as in Fig. 1, with the flight path in grey. The blue cross indicates the location of the stability discussion in Sect. 3.3. Times (in h) are since model initialisation (on 9 March at 12:00 UTC) + xx hrs ("xx" as specified in top left corner of each plot).



steps). The very weak GWs observed in the T21-run exist from the very first model time step, and no new GWs are forced in the following time steps. Clearly, the topography plays a significant role in GW generation. Are the two experiments hence an indication of direct orographic GW generation? Keeping in mind that Sect. 3.3 implicated the jet as the likely source, this hint to orography is a puzzling result. We therefore investigate the hypothesis that the orography is responsible for the GW excitation in an indirect way.

**4.2   CTL-run vs. T21-run: What is the difference?**

Which synoptic scale differences then arise from the reduced orography that could induce GW excitation? As argued in Sect. 3.3 the GWs are likely excited by out of geostrophic balanced flow. Therefore, we compare the cross-stream ageostrophic wind (calculated as in Sect. 2.3) for the two model runs. In Fig. 9, the cross-stream ageostrophic wind is shown where all three following conditions are met: the values of the cross-stream wind are greater than $7.5\,\mathrm{ms}^{-1}$, the total wind speed is greater
than $20\,\mathrm{ms}^{-1}$, and latitudes are lower than $80°$ N. As mentioned in Sect. 3.3, a critical value of $7.5\,\mathrm{ms}^{-1}$ is used to indicate when the jet can spontaneously radiate GWs.

The CTL-run, in Fig. 9, has large cross-stream ageostrophic wind regions. These cross-stream ageostrophic wind regions are an indication of an imbalance between the Coriolis and the pressure gradient force in the jet. Early after model initialisation (12 and 18 h — Fig. 9a and b, the CTL-run indicates large out of balance jet regions over the ocean. Figure 9b depicts the CTL-run
jet reaching cross-stream ageostrophic winds of $10\,\mathrm{ms}^{-1}$). Six hours later the CTL-run jet is unbalanced over the Greenland mainland (Fig. 9c). The greater the cross-stream ageostrophic wind is, the more unbalanced is the jet, and the more likely it spontaneously emits GWs (Zülicke and Peters, 2006; Mirzaei et al., 2014).

After each imbalance in the jet a GW response is seen 6 h later 2 km higher and downwind of the imbalance region (comparing Fig. 9a to c with e to g). This height and area offset is understandable as the GWs take time to propagate from 8 to 10 km
meanwhile drifting horizontally. Taking the mid-range ($\frac{\min+\max}{2}$) vertical group velocity of ray 0 (Sect. 3.3), the GW packet will propagate 2.7 km vertically in 6 h. Throughout all shown time steps the unbalanced region is associated with a GW field downwind.

The T21-run (Fig. 9i to p) shows a totally different picture. Firstly, the cross-stream ageostrophic wind indicate a smaller region and a more balanced jet. Small regions of 7.5 m/s (no 10 m/s region) is seen over the ocean in Fig. 9i and j. No cross-
stream ageostrophic wind is observed upstream or over the Greenland mainland at 24 h after model initialisation (Fig. 9k). Matching the more balanced jet, GWs are almost nonexistent in the T21-run.

Only during one time step was the T21-run jet more unbalanced than the CTL-run (Fig. 9d and l). At forecast hour 30 (at flight time) a large area of imbalance occurs below the north-westernmost part of the flight track (Fig. 9l). This imbalance area (at 350 hPa) is larger in the T21-run and indicated cross-stream ageostrophic wind values exceeding $10\,\mathrm{ms}^{-1}$. Six hours later,
the T21-run indicated more GWs than the previous time step (north of the flight track — Fig. 9p), and for the first time, a comparable (if not a greater) GW field to the CTL-run (Fig. 9h) was observed.





The CTL-run, in Fig. 9, has larger and more intense cross-stream ageostrophic wind regions when compared to the T21-run. Throughout all shown time steps the greater unbalanced region is rewarded with a greater GW field. Therefore, we assert that the GWs are directly caused by the increased out of balance within the jet.

## 315  4.3  CTL-run vs. T21-run: What causes the difference?

If the CTL-run has GWs, and the T21-run does not, then the difference between the two model runs must be the source of the GWs. By now we have established that the jet and its related imbalance is the cause of the GWs. We know that the orography played a role in the balance of the jet, but we are missing a puzzle piece connecting these two.

Comparing the two model runs, the jet location and shape remained similar. The centre of the low pressure (being stronger
in the T21-run) was displaced ≈5° westwards (Fig. 10). To find the missing link the difference in the model basic variables (U, V, temperature, pressure and relative vorticity) was calculated. Subtracting the wind speed and relative vorticity of the T21-run from the CTL-run produced an interesting dipole structure (Fig. 11). In Fig. 11(e – h) green (brown) demarcates an area where the CTL-run wind speed was faster (slower) than the T21-run. In order to investigate the origin of this difference it is convenient to calculate relative vorticity.

$$\zeta = \frac{\partial v}{\partial x} - \frac{\partial u}{\partial y} \tag{6}$$

It is well known that an uplift process induces relative vorticity (Holton, 2004). It then should be remembered that relative vorticity is only a different expression for the same wind field and that any process which causes changes of vorticity is actually altering the wind velocity field.[7] Indeed we find the same dipole structure which we observed in the wind velocity in the relative vorticity but offset by 90°. The dipole structure in the wind speed and vorticity represents the changes that occurred in the wind
speed, and consequently the changes that occurred in the jet.

The role of vorticity is known from the classical formation model of a Rossby wave (Holton, 2004). The flow above the ridge is compressed by the elevated orography, changing the potential temperature gradient, which in turn changes the vorticity, deflecting the wind and the synoptic flow. For the early time steps of 12hrs and 18hrs (Fig. 11a, b) there is a vorticity difference between the T21-run and the CTL-run over Greenland close to the coast. This is the same area where a change in uplift process would be expected due to the changes in the T21-run and CTL-run topography. This illustrates the effect that the uplift by orography has on the vorticity (and the jet). Later time steps (Fig. 11c – d and g – h) are more complex. We expect that the jet adjusted itself due to the lack of orography, for example; an adjustment to the orientation of the jet would additionally influence the relative vorticity field.

Vorticity can also be introduced by dissipation processes. This includes blocking (Smith (1982)), flow splitting (Siedersleben
and Gohm, 2016), mountain wakes (Grubisic, 2004; Siedersleben and Gohm, 2016; Smith, 2018), breaking GWs (Siedersleben and Gohm, 2016) and wakes at the edges of mountain ranges (Grubisic, 2004). Given the location and synoptic conditions, all

---

[7]This is because vorticity and wind velocity distribution are different views of the same wind field linked by a mathematical transformation and not because of a physical cause and effect.



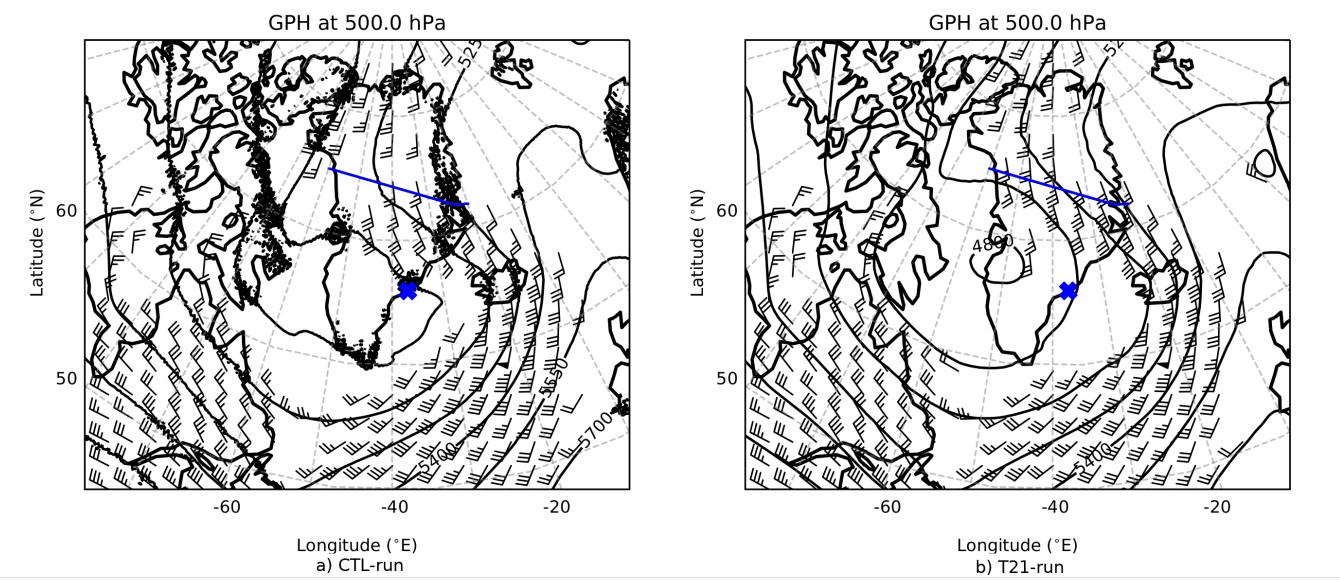

**Figure 10.** Geopotential heights and winds at $500\,\mathrm{hPa}$ valid for 9 March 18:00 UTC for the CTL-run (a) and the T21-run (b). Note the spotted (especially along the coast lines) CTL-run is a result of the orographic GW drag parameterization scheme. The rest of the display is the same as in Fig. 1, but with no PV line.

of the above processes most probably played a role in producing vorticity, but the dominant process (due to the altitude of the jet) is expected to be due to the compression of flow above Greenland.

We note the difference in orography from the CTL-run to the T21-run induced the different vorticity areas (Fig. 11). These

changes in the wind field will change the components in the jet, bringing the Coriolis and pressure gradient forces out of balance. This will trigger the jet to spontaneously emit GWs in order to bring the forces back into balance. Trüb and Davies (1995) showed in an idealised model simulation that evanescent GWs form over broad terrain in flow with a Rossby number (Eq. (7)) $< 0.25$. Also, upwind and downwind of the mountain a change in the wind components was observed. For our case, using a wind of $30\,\mathrm{ms}^{-1}$, Coriolis parameter of $0.00014\,\mathrm{1/s}$ and a mountain half-width measured from the blue X (Fig. 1) to

the Greenland north coast of $1650\,\mathrm{km}$ a Rossby number of 0.13 is achieved from

$$R_o = U/fL \qquad\qquad (7)$$

where $U$ is wind speed and $L$ is mountain half-width (the width of the mountain at $0.5 \cdot \mathrm{mountain\ height}$). The large evanescent GW (this should not be confused with the observations in Fig. 3; which clearly are propagating GWs), which is expected to form following Trüb and Davies (1995) over the Greenland terrain, can be one explanation of the rotation (Fig. 12)

and the upstream slow down of the wind. Wind being uplifted by orography will decrease due to kinetic energy changing into potential energy. In the geostrophic balance the Coriolis parameter is multiplied by the wind to obtain the Coriolis force, thus a

**Figure 11.** Difference between the CTL-run and T21-run for relative vorticity (left) and background total wind velocity (right). The revealed dipole structure is closely related to the GW excitation. Valid for 8 km and times similar to Fig. 9. The dipole structure in the wind speed difference (e–h) and in the relative vorticity difference (a–d) field is offset by half a phase (90°). The wind barbs are overlaid for wind speeds > 20 m s$^{-1}$, similar to Fig. 1, and the flight path is in grey





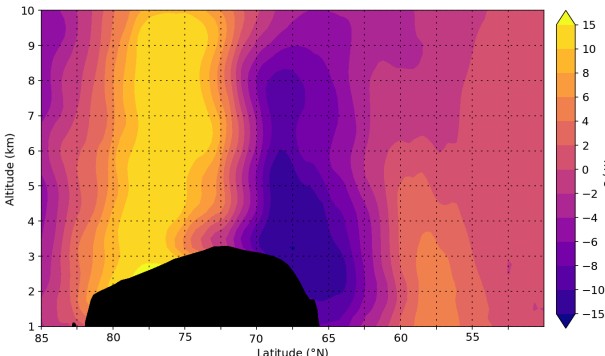

**Figure 12.** Difference between CTL-run and T21-run zonal (u) component wind at flight time. The zonal wind component represents an along-ridge wind (similar to Fig. 11d from Trüb and Davies (1995). The cross-section with altitude is aligned roughly along the jet axis (see Fig. 1 for the jet) at $40°$ W longitude from $85°$ N to $50°$ S. Greenland topography is indicated in black.

slow down in the wind will change the Coriolis force. The Coriolis force deflects winds to the right in the Northern Hemisphere, acting on the zonal (u) component of the wind. This explains the changes in the zonal component in Fig. 12. Figure 12 is a clear indication of the horizontal and vertical extent that orography has on the background wind.

## 5 Summary

This has been the first GLORIA limited angle tomography retrieval using an irregular (Delauney) grid and the Laplacian regularisation discussed in Krasauskas et al. (2019). Using GLORIA, on 10 March 2016 we observed GWs over Greenland in an area, where multiple possible GW sources exist. Possible GW sources included the jet or wind shear embedded in a breaking Rossby wave, or orography. Observations show the GWs to have a long horizontal wavelength ($\approx 330\,\mathrm{km}$) and short vertical wavelength ($\approx 2\,\mathrm{km}$). The temperature amplitude is $4.5\,\mathrm{K}$. The eastwards (upwind) tilt of the observed GW phase fronts (Fig. 3) indicates a vertically propagating GW. Using ERA5 reanalysis winds it is determined that this GW is upward propagating. Intrinsically the GW packet propagates horizontally against the wind at $25 - 72\,\mathrm{ms}^{-1}$ (groundbased velocity of 6 $- 17\,\mathrm{ms}^{-1}$). This is very fast compared to its vertical group velocity, which is $0.05 - 0.2\,\mathrm{ms}^{-1}$.

The GLORIA observed horizontal and vertical wavelengths as well as the calculated frequency were used as input into GROGRAT. Backtracing rays trace into the jet, with one ray (ray 3 in Fig. 4) descending to the Greenland plateau. In spite of the GWs drifting horizontally for 100's of kms with little to no vertical propagation, these GWs are vertically propagating GWs. Our study illustrates how far vertically propagating GWs can drift horizontally from their source. This reflects on the nonphysical nature of single column parameterization schemes currently in use for GWs.

The GROGRAT raytracing rays passed multiple regions where the jet was out of balance. Rays 0 to 2 passed above (and through - for ray 3) elevated values of the cross-stream ageostrophic wind over the mainland and through elevated values over the ocean (Figs. 5 and 6). The cross-stream ageostrophic wind is an indicator for an imbalance between the Coriolis and





pressure gradient forces in the jet exit region (Zülicke and Peters, 2006; Mirzaei et al., 2014). Such an imbalance in the jet exit region is normally brought into balance by spontaneous emission of GWs. Associated WKB violations were observed for ray 3 around 6.5 and 8 km (Fig. 6). This compares well to the hodographs which indicated downward propagating GWs between 7
and 10 km (Sect. 3.1).

For coherency, numerical experiments were designed. Two model runs were compared: one with the usual operational ECMWF forecast settings (CTL-run) and one with a flattened and smoothed orography (T21 topographic field — T21-run). All model runs produced similar meteorological fields, while the T21-run produced virtually no GWs (Fig. 9). On first glance, without further analysis, this would have formed a compelling (but incorrect) argument that the likely GW source would be a
typical case of mountain waves, i.e., a direct effect of orography.

Further investigation, however, revealed that changing the orography caused the cross-stream ageostrophic wind to differ between the model runs (Fig. 9). For all time steps, leading up to flight time, the CTL-run jet produced larger areas of imbalance and higher values in the cross-stream ageostrophic wind (except for time step 30 - flight time). The areas of greater imbalance (including time step 30, when the T21-run had a stronger imbalance region) were followed ≈6 h later by a stronger GW field.
The location of the cross-stream ageostrophic wind and synoptic conditions observed in our case is very much in agreement with a synoptic situation probable to release spontaneous GWs, as discussed, for example, by Uccellini and Koch (1987) as well as Plougonven and Zhang (2014). It is concluded that the jet, which depends heavily on the orography, is responsible for the observed GWs.

A jet is regarded as a localised source, in the sense that it releases a spectrum of GWs. Raytracing experiments show that a
variation of the initial conditions in the forward raytracing converge to the observed GW field. On the other hand, backward raytracing is highly sensitive to the launch conditions of the ray. This shows that the excited GW spectrum expands from the source and organises itself by the propagation conditions to GW packets of similar characteristics and spread over a large area. A similar behaviour (known as frequency-dispersion) is known for ocean waves (Holthuijsen, 2007).

Large-scale vorticity is used to illustrate the link connecting the orography to the change in jet balance. Subtracting the total
wind of the T21-run from the CTL-run produced a dipole structure (Fig. 11). A similar dipole structure (with a 90° phase shift) was obtained by subtracting the T21-run vorticity from the CTL-run relative vorticity.

A well-established link exists between orography and orography-induced vorticity changes. Vorticity is produced by the compression of air above orography (Holton, 2004). Moreover, vorticity can be produced by dissipation which can include blocking, flow splitting, wakes of mountains, GW breaking, or the edges of mountain ranges (Smith, 1982; Doyle and Shapiro,
1999; Grubisic, 2004; Siedersleben and Gohm, 2016; Smith, 2018). All of the above-mentioned processes are expected to be present during this case, but, only GW breaking and the compression of air has the capacity to directly deposit vorticity within the upper tropospheric jet. It is shown by difference fields that flow over broad terrain is directly responsible for large changes in the jet. These changes would bring the jet out of balance, triggering the release of GWs.

Based on the chain of arguments presented above we find that the observed GWs were excited by the jet, which was heavily
influenced by orography through large-scale vorticity, forced by flow over broad terrain. The connected changes of the wind field often occurred upwind of the orography.





According to Plougonven and Zhang (2014), our understanding of GWs from jets are still too inadequate to understand all the dynamics. With exception of the modelling study of Trüb and Davies (1995), we could not find literature directly connecting orography with the release of GWs which is not mountain waves. Trüb and Davies (1995) goes further in saying that observational evidence of GWs linked to orography induced ageostrophic imbalanced flow "will be difficult" to obtain. As we could find no observational studies to have observed GWs from this orography-jet combination, we believe this to be the first documented observational evidence of this mechanism.

A marginally in balance jet approaching orography is a common feature at mid- and high-latitudes. Therefore it is likely that this jet-orography interaction causes the jet to come out of balance on a frequent basis in many regions. Gravity wave generation by this jet-orography mechanism is capable of producing spontaneous adjustment regions over Greenland, Scandinavia, Antarctica, South America, New Zealand and others. In numerical weather prediction models most of these GWs would be resolved, but a large part of the spectrum would not be accounted for in climate models which need to be operated on a lower resolution for long-term runs. Parameterization schemes which could represent these GWs do not exist for such excitation processes. These GWs are also difficult to diagnose. Considering statistical studies of observed or model-resolved GWs, GWs excited by the suggested orography-jet mechanism could frequently be misinterpreted as classical mountain waves despite quite different characteristics. This article hence illustrates how challenging it is to disentangle the sources of GWs.





## Appendix A

**Table A1.** Spectral Windows used during retrieval

|   | Spectral Windows ($\mathrm{cm}^{-1}$) | Used to retrieve |
|---|---|---|
| 1 | 790.6250 - 791.8750 | Temperature |
| 2 | 791.8750 - 792.5000 | Temperature |
| 3 | 793.1250 - 793.7500 | $CCl_4$ |
| 4 | 794.3750 - 795.0000 | $CCl_4$ |
| 5 | 795.6250 - 796.2500 | $CCl_4$ |
| 6 | 796.8750 - 797.5000 | $CCl_4$ |
| 7 | 798.1250 - 799.3750 | $CCl_4$ |
| 8 | 883.7500 - 888.1250 | $HNO_3$ |
| 9 | 892.5000 - 896.2500 | $HNO_3$ |
| 10 | 900.0000 - 903.1250 | $HNO_3$ |
| 11 | 918.7500 - 923.1250 | $HNO_3$ |
| 12 | 956.8750 - 962.5000 | Temperature |
| 13 | 980.0000 - 984.3750 | Temperature, $O_3$ |
| 14 | 992.5000 - 997.5000 | Temperature, $O_3$ |
| 15 | 1000.6250 - 1006.2500 | Temperature, $O_3$ |
| 16 | 1010.0000 - 1014.3750 | Temperature, $O_3$ |

## Author contribution

MG performed figure production, all data analysis and write-up of the article. PP supervised the research and helped extensively with the paper. IK performed flight planning, initiated the idea of a Greenland paper and supported initial analysis. CZ assisted in many discussions and with identifying the out of balance jet regions. JU supervised the research, assisted in the GLORIA retrieval process and produced the level 1 dataset. ME contributed valuable knowledge and experience to the discussions. FF and MR obtained funding for the campaign. FF (and team) also managed the GLORIA sensor to obtain measurements and managed the data. MR also contributed to discussions on the paper. All authors contributed to revision of the paper and its figures.

## Competing interest

The authors declare they have no conflict in interest.



*Acknowledgements.* Jureca supercomputing facilities were used for the retrieval process. I would like to acknowledge the everyone that contributed to the campaign, especially the FX team and the pilots. Much of the article is based on the model runs produced by Inna

440 Polichtchouk, thank you for the model experiment and support. I would like to thank Andreas Dörnbrack for the many interesting discussions we had regarding the case. I also would like to thank E. Geldenhuys for her undying support.

**Financial support**

This work was partly funded by Deutsche Forschungsgemeinschaft (DFG) project PR 919/4-2 (MS-GWaves/SV), which is part of the DFG researchers group FOR 1898 (MS-GWaves). CZ received partial funding from the same research group for the

445 project Spontaneous Imbalance (ZU 120/2-2).





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
