# Peer review of "Orographically-induced Spontaneous Imbalance within the Jet Causing a Large Scale Gravity Wave Event"

_Atmospheric Chemistry and Physics, 2020_

## Author Response (AR1)

**Response to Reviewers comments on the preprint: Orographically-Induced Spontaneous Imbalance within the Jet Causing a Large Scale Gravity Wave Event**

Markus Geldenhuys[1,2], Peter Preusse[1], Isabell Krisch[3], Christoph Zülicke[4], Jörn Ungermann[1], Manfred Ern[1], Felix Friedl-Vallon[5], and Martin Riese[1]

[1]Forschungszentrum Jülich, Institute of Energy- and Climate Research, Stratosphere (IEK-7), Jülich, Germany
[2]South African Weather Service, Private Bag X097, Pretoria 0001, South Africa
[3]Deutsches Zentrum für Luft- und Raumfahrt, Institut für Physik der Atmosphäre, Oberpfaffenhofen, Germany
[4]Leibniz Institute of Atmospheric Physics, University of Rostock, Kühlungsborn, Germany
[5]Karlsruhe Institute of Technology, Institute of Meteorology and Climate Research - Atmospheric Trace Gases and Remote Sensing (IMK-ASF), Karlsruhe, Germany

**Correspondence:** M. Geldenhuys (m.geldenhuys@fz-juelich.de; markusgeld@gmail.com)

**1   Changes to manuscript unrelated to the reviewers' comments**

A minor addition have been made in Section 4.3 (CTL-run vs. T21-run: What causes the difference?). A new insight into Uccellini and Koch (1987) Eq. 9 has sparked the change. This change do not change the content nor the conclusions of the manuscript, however, the article would be more complete with this there-in.

**2   Response to Anonymous Referee 1**

We would like to thank the anonymous referee for reviewing our manuscript. The comments add value and overall improved the scientific value of the work.

**2.1   Response to Specific comments**

*l. 43: In stating that 'most' GCMs do not resolve gravity waves properly, the authors seem to indicate that some GCMs do resolve them well. If I understand the situation correctly, those GCMs are too expensive to be applicable for climate simulations, right?*

Yes, that is correct. I have added the following sentence in the manuscript to reflect this: "The few GCMs that do resolve a large spectrum of GWs are computationally too expensive for climate and chemistry runs."

*Section 2.4: What speaks against dividing the cross-stream ageostrophic wind speed by the total horizontal wind speed and using the resulting Lagrangian Rossby number as a measure of deviation from geostrophic equilibrium? This would look more intuitive to me, while the cross-stream ageostrophic wind speed only indicates imbalance when it is comparable to the total*

*horizontal wind speed. Moreover, many experts would not accept the identification of spontaneous imbalance with geostrophic adjustment (e.g. Plougonven Zhang 2014). The first is a true emission process, while the second is an initial-value problem. I would encourage the authors to keep these things better apart from each other.*

This is a good comment. Some of the following reasoning has been included in the manuscript. In Zülicke and Peters (2006) it is argued that the cross-stream ageostrophic wind velocity can equally serve to diagnose an unbalanced flow field - an idea which originates from quasi-geostrophic theory tracing back to Koch and Dorian (1988). Later, Mirzaei et al. (2014) use a threshold for the cross-stream ageostrophic wind speed. They go further in saying that this is comparable to using an ageostrophic Rossby number for selection of flow components which are faster than the Coriolis parameter ($f$). Further, their approach has less 'noise' than simply using the ageostrophic Rossby number. In our study, a similar more noisy dataset is obtained when using the cross-stream Lagrangian Rossby number.

The following has not included in the manuscript: Using ERA5 data to illustrate the effect in our study, Figures 1 and 2 (in this document) show similar results for the cross-stream ageostrophic wind and the cross-stream Lagrangian Rossby number. The cross-stream Lagrangian Rossby number covers a larger region and is noisier than the cross-stream ageostrophic wind, but both show unbalanced flow within the same region.

*Ls. 227-228 and table 1: Where in table 1 do I see a vertical wavelength? Or is this $lambda_y$? But then the caption would be incorrect.*

Thank you this slipped through! This has been corrected to $\lambda_z$.

*l. 269: How does a decrease in stability lead to a decrease in the vertical wavenumber? Is it not the other way round? At constant intrinsic phase velocity one would have N/m constant, with m the vertical wavenumber. This is for the mid-frequency range, but I would assume this is not changed substantially if the intrinsic frequency is close to f?*

A wave duct is formed when a more stable layer is sandwiched between two less stable layers. Wind speed and stability is known to be responsible for creating a wave duct. Although the wave duct is not strong enough to reflect the wave downwards (as discussed in Section 3.1 in the manuscript), a little energy will still be lost. This is what we wanted to touch on, but we see we did not bring this message across. However, you make a valid point that a decrease in stability leads to an increase in vertical wavelength. (I assume you meant vertical wavelength in your comment and not vertical wavenumber as is written. A decrease in stability will cause a decrease in vertical wavenumber.) Hence, I decided to avoid confusion and update the text accordingly.

*l. 10: the the is one the too much.* This has been resolved.

**3   Response to Anonymous Referee 2**

We would like to thank the anonymous referee for reviewing our manuscript. The comments were valuable and improved the work significantly.

[Figure]

**Figure 1.** Cross-stream ageostrophic wind calculated at 350hPa from ERA5 data. The top left panel starts on the 9th of March 2016 at 18:00 UTC and continues in a 6hrly timestep to 11th at 00:00 UTC. Data was only plot where the total wind speed was greater than $20\,\mathrm{ms}^{-1}$ and for cross-stream ageostrophic wind values less than $5\,\mathrm{ms}^{-1}$.

[Figure]

**Figure 2.** Same as for Fig. 1 for the variable Cross-stream Lagrangian Rossby number.

**3.1 Response to Major point**

*The authors have access in the simulations to all variables. The wave they are describing is a low frequency wave, as discussed for instance in the comments of the hodograph. A very good variable for capturing the signature and life cycle of this wave in the simulations would be the divergence of the horizontal velocity field. The signature of balanced motions is weak in this field, and waves with short vertical wavelengths come out conspicuously. The investigation of the divergence could in particular bring insights on the generation of the wave (in complement to the ray-tracing), and also importantly on whether traces of an analogous wave are present in the T21 run.*

The divergence field highlights the GWs upstream of the Greenland coastline and has been included in Figure 10 in the text. The divergence field did not highlight any stronger wave feature in the T21-run.

**3.2 Response to Specific comments**

*l26 Earths -> Earth's*

This has been addressed

*l30 for fronts, rather than a reference to a study of the parameterization of waves from fronts, reference to a study or to studies of the process itself, ie of emission by fronts, would be more appropriate. Here are some suggestions*

References for **??** have been added.

*l50-56: De la Camara et al 2016 also obtained improvements after modifications of their NOGWD scheme. It is mentionned at the end of Garcia et al 2017 that the improvement can be obtained by enhancing the NOGD too.*

As suggested, text from de la Camara et al. (2016) and Garcia et al. (2017) has been included in the paragraph.

*l76: is the acronym PGGS explained somewhere? Is there a reference describing it?*

The acronym PGGS is explained in line 76 (first submission). We have updated the text to make this clearer.

*l113-115: does the irregular grid have disadvantages? It is indicated that it allows to reduce the computation time. How important is this? Naively, one imagines that such calculation is done just once, so that computational expensiveness may be secondary, so long as it remains within reasonable bounds.*

Compared to previously used techniques, the approach used here showed no significant disadvantages to the other implementation (Krasauskas et al., 2019). Large 3D retrievals like the one presented in this paper require the use of supercomputers. Because of the nature of the problem, the calculation is done iteratively (and not only once). A typical tomographic retrieval will have 5 to 10 iterations. The run for this manuscript used $\approx 1\,000\,000$ points, this takes about 80 CPU hours with 6 processes and 2 threads. In practice, one needs dozens of these retrievals for a publication-quality 3D retrieval. Any computational cost savings are therefore relevant.

*l113-115: It is pointed out that this is the very first irregular grid retrieval. Is it possibel to quantify error bounds relative to the retrieval used usually?*

This submitted manuscript was the very first Limited Angle Tomography (LAT) Delauney method retrieval. **?** recently submitted a full-angle tomography Delaunay method retrieval. (Our newly submitted article will reflect the addition of LAT as opposed to the one submitted first.)

Regarding the comparison of the new retrieval methods to the old one: The publication Krasauskas et al. (2019) performed a detailed comparison of the new regularisation and Delaunay triangulation techniques with the previously used methods. Comparison of the new methods against the old ones are evaluated using synthetic data retrievals (Figure 3 in Krasauskas et al. (2019), compare rows A and D) and error bar comparison is shown in Figure 5, rows B and D. To summarise the paper, they found that the new method compared to the old method show similar structures within the tangent point area (high trust region).

*Some of uncertainties are discussed in lines 122-125; this is interesting. Perhaps a figure illustrating the sensitivity of retrievals to different choices could be shown in an appendix or as a supplementary material, so the interested reader may have an idea of the more robust features and the less reliable features of the retrieval.*

Robust features are indicated by the black line in Figure 3 of the manuscript. The black line indicate the volume with a high number of tangent points. Tangent points are the lowest point along the viewing trajectory, meaning its the densest and carries the most signal. The tangent point region has been found to be robust during tomography (Krisch et al., 2017, 2018). However the tangent point volume is rather thin during limited-angle tomography (as compared to full-angle tomography) and the surroundings are much less stable. Most of the retrieval experiments dealt in reducing obvious artefacts in the boundary regions outside the tangent point region. Reducing the artefacts make the cross-sections more visually pleasing and reduces the small, but given, impact on the tangent point volume. We use only data from this tangent point volume in the associated vertical cross-section plot and to the determine wave features in Table 1 of the manuscript. Meaning we only use the robust features. To avoid duplication the interested reader is directed to Krisch et al. (2018) for an in-depth discussion on the strengths and weaknesses of limited-angle tomography and the resultant effect on the reliability of features.

*l138: 'groundbased is too long for a subscript; a suggestion would be to write $\omega_{gd}$ and explain in the text that this corresponds to 'ground-based'. l138 Place the footnote after the word 'density' rather than after the parentheses which contains a mathematical formulation; there is no ambiguity because it would not really make sense to consider the fifth power of the expression... but still, it would be simpler to have the footnote after a word.*

Both of these were addressed.

*l139: 'taken into account': given the uncertainties on the damping due to turbulence or to the dissipation of waves in general, it would be worthwhile describing the assumptions used to account for these processes. Sensitivity to the choices made there could also be welcome, when results are presented.*

A reference is now provided which describes the damping in full.

*l142: what other indications of potential sources are there? Reaching the ground... For waves emanating from couvective regions, is the WKB condition violated?*

Within the raytracer, there are no other (other than the WKB parameter) indications that can be used to diagnose sources. GROGRAT relies on the interpretation of the user to realise the potential sources at hand and test each source. Very similar to

what was done in the rest of the manuscript. In the near future, it is the hope to implement more diagnostic tools to help with the source identification.

115    At this point in the manuscript, it is too early to discuss whether the WKB parameter was violated or not. This is discussed (and shown) later on in the manuscript (see Figure 6 and related discussions).

*l163: it would be useful to include a standard reference on the Savitsky-Golay filter, even if it seems classical.*

The classical paper that first introduced this filter has been added to the text.

*l173: the footnote is not well placed*

120    This was addressed.

*l182-183: 'the divergence of the jet': what exactly does this designate? This is ambiguous.* The sentence was updated to be more specific. Now it reads: "However, the divergence of the winds within the jet remains ..."

*l190: the waves observed have a wavelength 2 km. Should this read 'wavelengths < 4km' rather than '> 4km'? The following sentence causes confusion.*

125    Thanks for pointing to the typo. The typo was vertical wavelength (is now changed to horizontal wavelength).

*l219: 'despite the expected increase': on such short vertical scales the increase is not expected to appear clearly, relative to all other causes of variation; the observations do not scan a range much larger than the vertical wavelength...*

The "despite the expected increase of amplitude with a decrease in density" has been removed.

*l224-225: could this assertion be more physically justified? What foes this criterion correspond to, and what are the other*
130    *possible causes?*

We elaborate more on this now by referring to Equation 4 in the manuscript.

*l275: the beginning of the section could use a sentence of paragraph explaining the purpose of the simulations.*

The following 2 sentences were added to Section 4 to introduce the reason for the numerical experiment. "Originally designed as an attempt to entirely rule out topography as a source, a numerical experiment with strongly reduced topography was
135    designed. This yielded unexpected results implicating topography as a major contributor."

*Figure 11: more informative than the wind barbs given that the panels are small and that only a limited number of wind barbs can fit, the authors should consider plotting geopotential; the pressure level to which this corresponds should be indicated in the caption.*

The wind barbs have been replaced by pressure isolines at the respective level.

140    *About the summary: the summary is a bit abrupt, and acronyms (GLORIA, GROGRAT) are used directly. It depends on the editorial instructions, but it may be worth reintroducing them for hasty readers.*

We completely agree. The summary has been updated to be more 'standalone' to make more sense to hasty readers.

*l367: the horizontal phase speed is an important quantity, and the range given is very wide. The uncertainty in the estimate of this important quantity is worth a comment.*

145    The range given is not an uncertainty range, these are the values for rays 0 to 2 between 7.5 to 12.3 km. Because of the ray and altitude range, a range in the phase speed is obtained. The article has been updated to make it clear that these values are over a range of altitudes. The large changes in the phase speed with altitude is a result of the large changes in wind speed.

*l372-373: this is an important issue currently; reference to previous work highlighting this issue would be relevant, eg Sato et al (2012)*

150    Four references were added as an example.

*l380: 'For coherency'? Perhaps state explicitly the goal of the simulations, the hypothesis that is tested with these simulations.*

This has been addressed.

*l392: 'the jet, which depends heavily on the orography': the formulation is ambiguous... the dynamic of the jet is influenced*
155  *by the orography?*

This has been clarified in the text.

**References**

de la Camara, A., Lott, F., Jewtoukoff, V., Plougonven, R., and Hertzog, A.: On the gravity wave forcing during the southern stratospheric final warming in LMDZ, J. Atmos. Sci., 73, 3213–3226, https://doi.org/10.1175/JAS-D-15-0377.1, 2016.

160 Garcia, R. R., Smith, A. K., Kinnison, D. E., de la Camara, A., and Murphy, D. J.: Modification of the Gravity Wave Parameterization in the Whole Atmosphere Community Climate Model: Motivation and Results, J. Atmos. Sci., 74, 275–291, https://doi.org/10.1175/JAS-D-16-0104.1, 2017.

Koch, S. E. and Dorian, P. B.: A mesoscale gravity wave event observed during COOPE. Part III: Wave environment and probable source mechanisms., Mon. Weath. Rev., 116, 2570–2592, https://doi.org/10.1175/1520-0493(1988)116<2570:AMGWEO>2.0.CO;2, 1988.

165 Krasauskas, L., Ungermann, J., Ensmann, S., Krisch, I., Kretschmer, E., Preusse, P., and Riese, M.: 3-D tomographic limb sounder retrieval techniques: Irregular grids and Laplacian regularisation, Atmos. Meas. Tech., 12, 853–872, https://doi.org/10.5194/amt-12-853-2019, https://www.atmos-meas-tech.net/12/853/2019/, 2019.

Krisch, I., Preusse, P., Ungermann, J., Dörnbrack, A., Eckermann, S. D., Ern, M., Friedl-Vallon, F., Kaufmann, M., Oelhaf, H., Rapp, M., Strube, C., and Riese, M.: First tomographic observations of gravity waves by the infrared limb imager GLORIA, Atmos. Chem. Phys.,

170 17, 14 937–14 953, https://doi.org/10.5194/acp-17-14937-2017, 2017.

Krisch, I., Ungermann, J., Preusse, P., Kretschmer, E., and Riese, M.: Limited angle tomography of mesoscale gravity waves by the infrared limb-sounder GLORIA, Atmos. Meas. Tech., 11, 4327–4344, https://doi.org/10.5194/amt-11-4327-2018, https://www.atmos-meas-tech.net/11/4327/2018/, 2018.

Mirzaei, M., Zülicke, C., Mohebalhojeh, A., Ahmad-Givi, F., and Plougonven, R.: Structure, Energy and Parameterization of Inertia-Gravity

175 Waves in Dry and Moist Simulations of a Baroclinic Wave Life Cycle, J. Atmos. Sci., 71, 2390–2414, 2014.

Uccellini, L. W. and Koch, S. E.: The Synoptic Setting and Possible Energy Sources for Mesoscale Wave Disturbances, Mon. Weath. Rev., 115, 721–729, https://doi.org/10.1175/1520-0493(1987)115<0721:TSSAPE>2.0.CO;2, 1987.

Zülicke, C. and Peters, D.: Simulation of inertia-gravity waves in a poleward-breaking Rossby wave, J. Atmos. Sci., 63, 3253–3276, https://doi.org/10.1175/JAS3805.1, 2006.